# p38γ Activation and BGP (Biliary Glycoprotein) Induction in Primates at Risk for Inflammatory Bowel Disease and Colorectal Cancer—A Comparative Study with Humans

**DOI:** 10.3390/vaccines8040720

**Published:** 2020-12-02

**Authors:** Harvinder Talwar, Benita McVicker, Martin Tobi

**Affiliations:** 1Research and Development VA Medical Center and Internal Medicine, Wayne State University, Detroit, MI 48201, USA; ar8673@wayne.edu; 2Research Service, VA Nebraska-Western Iowa Health Care System, The University of Nebraska Medical Center, Omaha, NE 68105, USA; bmcvicker@unmc.edu; 3Research and Development Service, Department of Internal Medicine, Detroit VAMC, Detroit, MI 48201, USA; 4Central Michigan University College of Medicine, Mount Pleasant, MI 48859, USA

**Keywords:** CRC, p38γ, pp38γ, BGP, CEA, blood group antibodies, p87

## Abstract

Colorectal cancer (CRC) is a common cause of cancer-related deaths largely due to CRC liver metastasis (CRLM). Identification of targetable mechanisms continues and includes investigations into the role of inflammatory pathways. Of interest, MAPK is aberrantly expressed in CRC patients, yet the activation status is not defined. The present study assessed p38γ activation in CRC patients, cancer cells, and tissues of cotton top tamarin (CTT) and common marmoset (CM). The primate world is an overlooked resource as colitis-CRC-prone CTT are usually inure to liver metastasis while CM develop colitis but not CRC. The results demonstrate that p38γ protein and phosphorylation levels are significantly increased in CRC patients compared to normal subjects and CTT. Furthermore, p38γ phosphorylation is significantly elevated in human CRC cells and hepatoblastoma cells but not in CM colon. Additionally, carcinoembryonic antigen (CEA) and biliary glycoprotein (BGP) are induced in the CRC patients that showed p38γ phosphorylation. Inhibition of p38 MAPK in CRC cells showed a significant decline in cell growth with no effect on apoptosis or BGP level. Overall, p38γ is activated in CRC tumorigenesis and likely involves CEA antigens during CRLM in humans but not in the CTT or CM, that rarely develop CRLM.

## 1. Introduction

Colorectal cancer (CRC) is the third most common cancer with respect to global mortality [1]. The majority of deaths are due to the development of colorectal liver metastasis (CRLM) as the liver is the most ominous site of distant metastatic spread in CRC patients, often heralding demise [2]. Despite advancements in surgical interventions and chemotherapeutics, CRLM morbidity is a leading healthcare concern emphasizing the need to define contributing mechanisms. The role of signaling cascades in CRC tumor cell progression is an avid area of investigation with associated factors implicated as potential mechanisms [3,4,5,6]. In previous work of an animal preclinical model contrasted with humans, we found a structural translocation in the distribution of a downstream VEGF-effector, biliary glycoprotein (BGP-CEACAM-1), in livers bearing metastasis in the human [7]. This suggested a paraneoplastic hepatopathy but patients with all-cause cirrhosis and hepatitis did not demonstrate this translocation. As part of the current effort to define contributing mechanisms, our goal was to investigate the role of carcinoembryonic antigen molecular family (CEACAMs) and MAPK signaling in CRC patients, cancer cells, and primate colon tissue. Additionally, we hypothesized and assessed the relationship of prometastatic blood group antigens [8] and expression of a putative innate immune system glycoprotein 87 (p87) [9] to that of BGP in a preclinical model.

CEA (CEACAM5) and biliary glycoprotein (BGP) belong to the immunoglobulin adhesion molecule superfamily and have been implicated in colon tumorigenesis and metastasis [10]. CEA and BGP are overexpressed in cells of the majority of colorectal carcinomas and this is associated with measurable serum levels, recurrence rates, and liver metastasis [11]. Studies have shown that a direct relationship exists between CEA and the metastatic potential of CRC cells [12]. CEA stimulates tumorigenic factors produced by hepatic cells that are prometastatic. BGP also has been implicated in tumor development and survival [13,14]. However, it remains to be determined what role BGP, CEA, and related signaling mechanisms have in CRC proliferation and tumorigenicity.

Along with CEACAMs, MAPK signaling and p38γ activation have been implicated in colon tumorigenesis [15,16,17,18]. Studies have shown p38γ and p38δ activation associates with inflammation-related colon cancer development [15,18]. Further, p38 MAPK signaling is suggested as a contributor in obesity-related CRC [19]. In addition, a recent study by Tomas-Loba et al. showed a role of enhanced p38γ expression during intrahepatic hepatocellular carcinoma (HCC) metastases [20]. However, the contribution of p38γ signaling in colorectal cancer liver metastasis is not defined.

The present study was undertaken to determine the role of CEACAMs and p38γ in CRC patients compared with normal subjects, CRC cell lines (HCT 116, HT-29, Caco-2, and SW620), and a liver hepatoblastoma cell line (HepG2). In addition, we studied p38γ and other liver metastasis-associated antigens in the cotton-top tamarin (CTT), a spontaneously occurring primate model for colitis and cancer [7,21] that does not usually develop liver metastasis after contracting CRC. We contrasted the latter to its close cousin, the common marmoset (*Callithrix jacchus*) which develops colitis but not CRC and compared p38γ and adhesion molecule expression. We further investigated the effects of p38 MAPK inhibitors [22,23] on tumor cell growth and apoptosis in CRC cancer cells. The results demonstrate that p38γ protein and phosphorylated p38γ are significantly increased in CRC patients and that phosphorylated p38γ is induced in CRC tumor cells. Moreover, inhibition of p38 MAPK signaling abrogates cell growth without affecting apoptotic cell markers in the CRC cells. Additionally, BGP and CEA protein levels are similarly increased in CRC patient tissues. The relationship of BGP to hepatic metastasis as indicated previously [7] was augmented herein with positive correlations shown for both prometastatic antigens and prognostic biomarker p87 [9], a product of gut Paneth cells known as key effectors of the innate immune system.

Overall, this work indicates that p38γ is activated in CRC tumorigenesis and likely involves CEA antigens during CRLM in humans but not in the CTT or CM, that usually do not develop CRLM. Studies using the CTT model that develops CRC but rarely liver metastasis has led to several suggested mechanisms of avoidance of CRLM. Most of the putative mechanisms involve changes in the immunoglobulin superfamily of adhesion molecules but there is little understanding of the role of inflammatory markers and signaling cascades. This report contributes new information concerning MAPK activation and association with prometastatic antigens. Based on these findings, vaccines or decoy nanoparticles and vaccines may be devised to obviate CRLM in the human.

## 2. Materials and Methods

### 2.1. Chemicals and Antibodies

Phospho-p38γ antibody was a generous gift from Dr. J Luo [24]. Doxorubicin treated cells were a generous gift from Dr. A. Rishi (26). CEA (T84.66) antibodies were the generous gifts received from Dr. J. Shively and Dr. S Hefta (City of Hope, Duarte, CA, USA) and the Tosoh Medics quantitative CEA testing was performed by Lois Nakayama. 4D1/C2 anti-BGP monoclonal antibody was provided from Dr. C. Wagener (Freiberg, Germany); anti-CEA family C53.5 and anti-CEA 46.1 was provided by Dr. WJ Allard and from Bayer Diagnostics (Tarrytown, NY, USA). The FH6 monoclonal antibodies to fucoganglioside 6B, FH2 to sialylated Lewis X and SH1 to extended Lewis Y blood groups were gifted by Dr. Hakomori (University of Washington, Seattle, WA, USA) [8]. Anti α-sialyl Tn monoclonal was gifted by Dr. ME Key (DakoCytomation Inc., Carpinteria, CA, USA); and CaCo3/61 against fucosylated aminoproteoglycans blood groups was a generous gift from Dr. A. Quaroni (SUNY, Ithaca, NY, USA). Total p38 and p38γ antibodies were purchased from Cell Signaling Technology (Beverly, MA, USA). Horseradish peroxidase (HRP)-conjugated anti-mouse and anti-rabbit IgG secondary antibodies were obtained from Cell Signaling Technology (Danvers, MA, USA). MAPK inhibitors, Losmapimod (inhibitor of p38α and β) and Doramapimod (inhibitor of p38α, β, γ, and δ) were from Cayman Chemicals (Ann Arbor, MI, USA). Solid-phase ELISA was performed as described previously [9,25] and results were expressed as OD-background at protein concentration of 5 µg protein/well. Seeing that rodents are commonly used in cancer research we amplified the applicability of our data by determining total p38 and p38γ in rat liver and in small bowel rat IEC-6 cells. Since equivalent human small bowel mucosal cells lines are not generally available, this provided a convenient alternative. In order to determine that the CTT likely expresses total p38 we performed a BLAST analysis on the protein expression of the closely related CM using the published genome and verified the presence of p38.

### 2.2. Cell Lines, Culture Conditions, and Viability Measures

Human cancer cell lines (HT-29, SW620, Caco-2, HCT-116, and HEPG2) and rat small bowel epithelial IEC-6 cells were purchased from ATCC. All cells were cultured and maintained in DMEM or Eagle’s minimal essential medium, containing 10% fetal bovine serum and 1% penicillin and streptomycin at 37 °C with 5% CO_2_/95% air. Cells used in all experiments were at 3–4 passages and synchronized for growth phase. The basic growth media and antibiotics were obtained from Invitrogen (Waltham, MA, USA). Fetal bovine serum (FBS) additive was purchased from Denville Scientific Inc. (Metuchen, NJ, USA), and DMSO was purchased from Thermo Fisher Scientific (Fair Lawn, NJ, USA).

Cell viability was measured using the MTT (3-(4, 5)-dimethyl thiazol-2, 5-diphenyl tetrazolium bromide) assay as described [26]. HT-29 and HCT-116 cells were seeded in 96-well plates and incubated with media alone (untreated control), or treated with Losmapimod and Doramapimod for 24, 48, and 72 h, respectively. After treatment, MTT reagent was added at 0.5 mg/mL concentration for 2–4 h at 37 °C. DMSO was added to solubilize formazan and the plate was read at 570 nm.

### 2.3. Tissue Sources, Protein Extraction, and Western Immunoblotting

Tissues from normal subjects, CRC patients, CTT, and common marmoset were obtained by colonoscopic biopsy and animal necropsy from MARCOR (Marmoset Colony of Oak Ridge, University of Tennessee at Oak Ridge, Oak Ridge, TN, USA) and the NEPC (New England Primate Center, Southborough, MA, USA). Specimens were collected subject to protocols approved by the ethics committees of the institutions involved, and informed consent was obtained from the patients where applicable. A membrane-enriched tissue extract (MEE) was prepared from the biopsies by homogenization in RIPA buffer containing protease and antiphosphatase inhibitors and clarified by centrifugation (1000× *g*) and the supernatant sonicated and centrifuged (10,000× *g*) [27]. Cell extracts from various cancer cell lines, after the aforementioned treatments, were made in RIPA buffer containing protease and phosphatase inhibitors. Equal amounts of proteins (20 µg) were mixed with the same volume of 2X sample buffer, separated on 10% SDS-polyacrylamide gel electrophoresis and transferred to a polyvinylidene difluoride (PVDF) membrane (Bio-Rad, Hercules, CA, USA). The PVDF membrane was blocked with 5% dry milk in TBST (Tris-buffered saline with 0.1% Tween-20), rinsed, and incubated with primary antibody overnight. The blots were washed and incubated with a HRP-conjugated secondary anti-IgG antibody. Membranes were washed and immunoreactive bands were visualized using a chemiluminescent substrate (ECL-Plus, GE Healthcare, Pittsburgh, PA, USA). Images were captured on Hyblot CL film (Denville Scientific Inc., Metuchen, NJ, USA). Optical density analysis of signals was performed using Image Quant software (version 5, GE Healthcare, Chicago, IL, USA) [26].

### 2.4. Statistics

Differences of means were analyzed by the parametric Student’s *t*-test using an online statistical program (www.vassarstats.net). Nonparametric tests (chi-square, Fishers, Pearson) were used to analyze proportions and the linear correlation analysis by least squares method was used to obtain correlation coefficients (r) and test for significance. *p* values were considered significant at the *p* < 0.05 level.

## 3. Results

### 3.1. Increased p38γ and pp38γ in CRC Patients Compared to Normal Subjects

Previous studies have shown the overexpression of p38γ in CRC subjects by immunohistochemistry [16,27,28]. Recent studies have shown that p38γ is required for inflammation-associated colon tumorigenesis [18] and p38γ is essential for intrahepatic HCC metastases [20]. Our aim was to study the activation (increased phosphorylation) of p38γ in normal and CRC patients. We determined both p38γ and pp38γ levels in CRC patients and normal subjects by Western blot analysis. The results in Figure 1a clearly demonstrate that p38γ (2.5 fold) and pp38γ (2.9 fold) levels are significantly increased in CRC patients. β-actin band is included for equitable loading. Figure 1b,c represents the densitometry values (mean ± SEM) of the phosphorylated form of p38γ and p38γ from 10 CRC patients and 10 normal subjects normalized to total p38.

### 3.2. Presence of p38γ Protein and pp38γ in Preclinical Tissues Compared to Human Colon Cancer Specimens

The results in Figure 2a show the presence of p38γ protein in preclinical models (normal human tissue and CTT liver, lanes 1 and 4 respectively) in comparison with CRC patients with pp38γ detected only in the human cancerous tissues (lanes 2 and 3). Furthermore, we determined the presence of p38γ protein in the common marmoset (CM-*Callithrix jacchus*) colon tissues. Figure 2c shows that p38γ protein (46 kDa) is not detected by p38γ antibody, whereas total p38 (α and β isoforms) is present in all the samples. It is an important aim to illustrate clearly that tissues susceptible to cancer should express p38γ. We also determined the presence of p38γ in different organs of common marmoset using specific p38γ antibody. The results in Figure 2d show that p38γ protein is expressed in the liver (lanes 1 and 2), but it is not detected in either gall bladder or colon tissues. On the other hand, p38γ protein is strongly expressed in ileum (lane 5).

### 3.3. Increased Carcinoembryonic Antigen (CEA) and Biliary Glycoprotein (BGP) in CRC Patients

Previous study has demonstrated that mRNA levels and the concentration of CEA are significantly induced in CRC patients compared to normal subjects [29]. On the other hand, BGP mRNA levels are reduced in CRC patients [30]. Conversely, it is also reported that BGP levels are overexpressed in human CRC cells that have high metastatic potential [13,14]. Based on these results, we determined the CEA and BGP protein levels in the same CRC patients and normal subjects that were used for the determination of p38γ levels. The results in Figure 3a show that CEA and BGP proteins are induced in colon cancer patients compared to normal subjects. Figure 3b,c shows the densitometry values for CEA (3.15 fold) and BGP (6.3 fold) from 10 CRC patients and 10 normal subjects normalized to β-actin.

### 3.4. Increased pp38γ and CEA Protein Levels in CRC Cell Lines

We determined the phosphorylation levels of p38γ in CRC cell lines (HCT-116, HT-29, Caco-2, and SW620) and compared these with IEC-6 (normal rat intestinal epithelial cells) and HEPG2 (hepatocellular carcinoma cells). Figure 4a depicts phosphorylation levels that are significantly induced in all the CRC cell lines as well as in HepG2 (human hepatoblastoma carcinoma cell line) cell line compared to that of the IEC-6 cells. Similarly, Figure 4c shows that CEA levels are also increased in all CRC cell lines and HepG2 cell line when compared to IEC-6 cells.

### 3.5. Losmapimod (LOS) and Doramapimod (DORA) Inhibition of CRC Cell Growth

Previous studies have demonstrated that inhibitors of p38α and p38β, (SB203580 and other pyridinyl imidazole compounds) inhibited the cell growth of CRC cell lines [31,32]. In this study, we investigated the growth inhibitory effects of losmapimod (inhibitor of p38α and β) [23] and doramapimod (inhibitor of p38α, β, γ, and δ) [22], on the HCT 116 and HT-29 CRC cell lines. Figure 5a,b shows the dose dependent cell growth inhibition by LOS and DORA in HCT 116 cells between 48 and 72 h of treatment. With respect to cell viability, ≈50–60% inhibition was observed between 48 and 72 h. Similarly, Figure 5c,d shows the similar inhibition of cell growth by LOS and DORA in HT-29 cells. The cell growth inhibition was statistically significant in both cell lines (*p* < 0.005).

### 3.6. Inhibition of p38 MAPK by Losmapimod and Doramapimod Has no Effect on Caspase 8 and 3 Activities

Activation of caspases has been regarded as a hallmark of cell apoptosis. To assess the effects of p38 MAPK inhibition by losmapimod and doramapimod, HT-29 and HCT 116 cells were pretreated with LOS and DORA for 24 and 48 h. The results in Figure 6 demonstrate that caspase 8 caspase 3 protein levels were not altered by p38 MAPK inhibition by losmapimod or DORA. The results further show that no cleaved products of caspase 3 were formed by p38 MAPK inhibition in both the cell lines.

### 3.7. Inhibition of p38 MAPK by Losmapimod and Doramapimod Has no Effect on BGP Levels

BGP is a member of the immunoglobulin gene superfamily [10]. Here, we investigated p38 MAPK inhibition by LOS and DORA on BGP levels in HT-29 and HCT 116 cells. Both cell lines were pretreated with LOS and DORA (5 and 10 µM) for 48 h. The results in Figure 7 show that p38 MAPK inhibition by losmapimod or doramapimod had no effect on BGP levels in both the cell lines.

### 3.8. Expression of Tumor Metastasis-Associated Antigens in Colon Mucosal Extracts of the Cotton Top Tamarin and Common Marmoset (Callitrichidae)

Figure 8 shows that BGP, CEA family members (other than T84.66 which is most specific to human CEA shown in lane 4) and p87 are all expressed equally in the Callitrichidae family of nonhuman primates (CTT and CM). There was insufficient material to assay CM tissue for reactivity to certain blood group antigens and these were only assayed in the CTT tissues and most found to be absent. The sialylated antigens are the most depressed as a group (lanes 5–8) perhaps explaining the lack of liver metastasis in these animals as these antigens usually correlate highly with liver metastasis [8]. Despite detectable CEA and BGP antigens in the colonic tissues our past work suggests that there are significant changes in Callitrichidae CEA, CEA liver receptors, and glycosylation that makes the liver an “infertile field” [7]. In addition, BGP is not expressed in the CTT liver (but is found in the gallbladder wall) [33]. BGP, a downstream effector of VEGF, clearly has detectable tissue levels in the colon tissue but apparently absent in the liver depriving metastatic cells of crucial angiogenic support. We therefore contrasted BGP with the expression of other tumor metastasis-associated antigens as determined by ELISA in CTT and CM colon tissue extracts.

### 3.9. Correlation of BGP with other Prometastatic Antigens in Preclinical Primate Models

Figure 9 shows a significant correlation of Adnab-9 expression (recognizing p87 a Paneth cell product) with BGP in CTT and CM. This suggests a relationship with the innate immune system which is the major defense against cancers and pathogens. There was also a significant correlation between BGP and CaCo3/61 expression (r = 0.96; *p* < 0.01) but only in CM tissues. This is interesting as immunohistochemistry with this antibody in CTT was positive in the colon [35]. This data is confirmatory of a previous study which found similar profiles in humans with inflammatory bowel disease [34]. Additionally, the scatter gram results shown in Figure 10 indicate a statistically significant correlation between the Adnab-9 and T84.66 antibody binding in tissues extracts from the CTT in both cancerous and normal material. T84.66 is a monoclonal antibody that binds with high specificity and affinity to human tumor-associated CEA. This suggests activation of the immune system in the face of inflammation and cancer and the prometastatic CEA prognostic marker. This is accurate in humans and that observation would probably apply to the CTT despite the fact that metastases are not usually found in the liver. The CM colonic tissues extracts did not show this correlation as they have very limited cancer risk as compared to CTT. All possible permutations from differences by Student’s *t*-test between CTT and CM tissues were not statistically significant. The relationship between BGP and CEA in CTT normal mucosal extracts are however, highly inversely correlated as shown in Figure 11, suggesting opposing but interrelated functions.

## 4. Discussion

Colorectal cancer (CRC) is a leading cause of cancer mortality with the majority of deaths associated with colorectal liver metastasis (CRLM). Ongoing research efforts are focused on the identification of targetable mechanisms to combat progressive CRC. A variety of preclinical models along with human CRC patient samples are employed in the search for potential therapeutic targets.

One overlooked resource is that the primate world is usually inure to advanced CRC and liver metastasis. CRC has been documented in new and old-world primates but the overall prevalence is unknown and unlike the CTT and CM the biological context of inflammatory bowel disease is undefined. Previous work in our laboratory using the CTT, which develops colitis complicated by CRC but rarely CRLM, suggests that five mechanisms of avoidance of CRLM may explain this phenomenon and be potential targetable mechanisms. Most of these putative mechanisms involve changes in the immunoglobulin superfamily of adhesion molecules (IgSFAM) and little work has been done as yet with other factors which can play a role in cancer. Of interest, the expression of MAPK is aberrantly expressed in CRC patients, yet the activation status is not defined. The present study assessed p38γ activation in CRC patients, CRC and liver cancer cells, and tissues of the CTT and CM, which develops colitis that is not complicated by CRC.

Efforts towards the identification of putative mechanisms of CRC and liver metastatic disease include the study of MAPK signaling and p38γ activation in particular. p38 MAPK was first identified and cloned in response to stress-induced cytokine expression [27,36]. Thus far, p38 MAPKs consist of four characterized isoforms (α, β, γ, and δ) and are mainly proinflammatory cellular kinases [37]. All MAPKs are activated by phosphorylation of threonine and tyrosine residues within Thr-Xaa-Tyr motifs without significant changes in their expression [38,39]. Studies indicate that p38γ (gene name: MAPK12) plays a significant role in colon tumorigenesis. First, p38γ is increased by the RAS oncogene that in turn induces Ras which engenders invasive activity [17,40,41]. Second, p38γ levels are expressed in several human malignancies including CRC [16,17], breast cancer [42,43], aside from the aforementioned hepatocellular cancer [20]. Studies have shown p38γ and p38δ activation links inflammation-associated colon cancer development [15,18]. Interestingly, a recent study by Tomas-Loba et al. has shown high expression of p38γ in human hepatocellular carcinoma (HCC) which is essential for intrahepatic HCC metastases [20]. Moreover, the procarcinogenic roles of leptin and associated liver diseases (i.e., NAFLD) have been linked to the promotion of CRC cell invasiveness via the activation of the MAPK signaling pathway [44]. Considering this, we hypothesized that the activation of p38γ could be a significant mediator involved in CRC tumorigenesis. Indeed, the results of this study demonstrate for the first time that the phosphorylation of p38γ is significantly increased in CRC patients. Although previous studies [16,28] have shown that p38γ protein is increased in CRC patients by immunohistochemistry, the immunoblot analyses of this work clearly demonstrate that p38γ protein and its phosphorylation product are strongly expressed in human colorectal cancers.

Our analyses were strengthened by the comparison of human CRC p38γ to preclinical models including the CTT. The CTT is known as a natural animal model for inflammatory bowel disease complicated by CRC [9,33,45]. We used the CTT liver as a pp38γ protein negative control as it evades hepatic metastasis, which is dependent on p38γ phosphorylation, at least in the HCC scenario [20]. In contrast, the close cousin of CTT, the CM, develops colitis but does not share the vulnerability of the CTT for CRC [46] and shows absence of p38γ and pp38γ in colon (Figure 2c). In contrast to the colon, the liver extract is p38γ positive in the CM (Figure 2d). Of interest also, is the finding of p38γ in the ileum of the CM. This is important as it suggests cancer susceptibility in the small bowel and the commonest cancer of elderly CM is adenocarcinoma of the small bowel [47]. The CTT is an endangered species and access to tissues are severely limited, which does not currently allow p38γ for comparisons in colonic tissue. Overall, we have posited several potential mechanisms by which the CTT evades liver metastasis [7,33] and the absence of hepatic pp38γ may be an additional novel mechanism. Given the predilection of humans for liver spread of CRC, our results showing enhanced phosphorylation of p38γ in CRC patients suggest that p38γ is a prometastatic protein in CRC.

In addition to the activation of p38γ, this study demonstrates that CEA and BGP proteins levels are significantly increased in the same CRC patients that have high p38γ protein. Similarly, the phosphorylation of p38γ and CEA levels are significantly induced in several colon cancer cell lines and HepG2 cell line compared to normal epithelium cells. Our results in the CTT and CM suggest a relationship between BGP and p87 expression and a relative lack of prohepatic metastatic blood group substance expression relative to BGP. These findings suggest some constitutive defense against both invading cancer and liver metastasis. This notwithstanding, one third of CTT die of CRC, so in contradistinction to liver metastasis, primary colon cancer poses a substantial threat to the survival of this endangered species.

The use of inhibitors to p38 that have variable specificities demonstrated that both losmapimod (p38 α and β inhibitor) and doramapimod (p38 α, β, γ, and δ inhibitor) significantly inhibited the cell growth of CRC cell lines. The inhibition was not uniform, with doramapimod slightly higher than losmapimod. We also investigated the effect of perfenidone, inhibitor of p38γ on the growth of HT-29 and HCT-116 cells. The results indicate that perfenidone treatment at two different concentrations (5 and 10 µM) inhibited only 20% of cell growth of both the cell lines (data not shown). On the other hand, losmapimod and doramapimod treatment of HT-29 and HCT-116 cells did not activate caspase 8 and caspase 3 indicating that interfering with the p38 MAPK pathway has no effect on the apoptotic death in these cell lines.

The role of p38 MAPK in regulating BGP protein levels was clarified in the CRC cells as no effect on BGP protein was observed following p38 MAPK inhibition. In comparison to the CTT that do not express BGP in liver tissues, the lack of effect of the p38 MAPK inhibition suggests that BGP has a separate role in human liver metastasis which remains to be fully defined. Since BGP is a downstream effector for VEGF and important in establishing a vital blood supply for liver metastases [33], we postulate that as long as BGP remains a structural biliary canalicular component it is not able to act as a VEGF effector. However, we have shown that BGP does translocate to the hepatocyte cytoplasm in patients with metastasis, thus allowing it to potentially promote liver metastases [7]. The question of whether this translocation is part of a human paraneoplastic hepatopathy that benefits the growth and spread of CRC is an alluring hypothesis and it is likely that BGP and p38 interact in this process. The finding of an inverse relationship between CEA and BGP is a novel finding in the CTT however, a seminal paper on familial adenomatous polyposis in humans compared the expression profiles of these two antigens in neoplastic tissue of varying dedifferentiation and found twofold decrease in BGP expression in the more differentiated mucosa whereas CEA levels were only 30% higher than normal mucosa in the equivalent tissue [48]. Furthermore, in a study of well-differentiated human lung adenocarcinoma A549 cell line, researchers found significantly diminished BGP (about seven times less) than CEA expression in the conditioned media of confluent cell growth by ELISA [49]. Conversely, the positive correlation between BGP and p87 may reflect a protective effect against neoplastic progression in the mucosa at risk.

To summarize, we demonstrate for the first time that p38γ activation (phosphorylation) is increased in CRC patients and CRC cell lines along with enhanced BGP levels in the CRC patients. p38 MAPK specific inhibitors (losmapimod and doramapimod) inhibited the cell growth without affecting the apoptotic cell death in colon cancer cells. Furthermore, BGP which is structurally similar to CEA, is not affected by p38 MAPK inhibitors in CRC cells. The CEA molecule is widely accepted as a colonic tumor marker that is significantly increased in CRC patients and used as an indicator of liver metastases [50,51]. The role of CEA in metastasis is not fully characterized but implicated as a prometastatic adhesion and signaling molecule that interacts with cells in the hepatic microenvironment [52,53]. Moreover, the prometastatic effects of CEA are enhanced in the setting of pre-existing liver disease including alcoholic [5,54] and nonalcoholic liver disease [55,56]. Most recently, it was reported that NAFLD and CEA expression are associated with an increased incidence of liver metastasis in CRC patients [55]. Thus, CEA and its family members having similar DNA sequence and structure homology such as BGP [13,14] might play a role in the metastatic process. Whether pp38γ MAPK is involved in enhanced CRLM is not known. However, the results of this study support the role of this MAPK in liver metastasis as we detected an increased pp38γ activation in CRC patients along with a lack of pp38γ in animals that do not develop CRLM. Indeed, the absence of pp38γ in extracts of CTT liver and common marmoset may provide a potential 6th mechanism by which the CTT evades liver metastasis [7,33]. A possible 7th mechanism may be the lack of blood group substance expression of the sialylated phenotype. We also postulate that differences in p38γ expression may explain why CTT contracts colitis complicated by cancer (p38γ positive) while CM does not (p38γ negative). Further study is necessary to substantiate this postulate. Our future aim is to better define the role of p38 and BGP, a family member of CEA, in colon-induced liver metastasis. Both the CTT and more recently the CM have been used as models for vaccination research, superior to the older macaque model [57,58]. By focusing on elements of the MAPK and an innate-immune system effector cell product we believe that we have provided a novel perspective that combines these elements that could be integrated into the newly advocated holistic approach to vaccine development [59]. Taken together, this supports an approach of using vaccination against prometastatic elements to avoid CRLM. In addition, since the CM is an animal model for serious coronavirus infection [60], this may have research implications for COVID-19 vaccine and therapeutics development. Given these considerations, and that the MAPK are active in innate cell immunity [61], we believe that our research is most relevant and may potentially inform future development in preventative strategies as has been done in rats using stem cell vaccination [62] based on the “seed” hypothesis that could be applied to the “soil” counterpart based on the above findings.

## 5. Conclusions

In this study, we directly assayed for the expression of phosphorylated p38γ in clinical, preclinical, and cell line preparations. Notably, pp38γ is absent in a preclinical model which has little predilection for liver metastasis. Clinical CRC specimens and cell lines do show elevated expression along with elevated BGP and CEA, unaffected by p38γ inhibitors. The relationship between BGP and p87 in the CTT and CM is intriguing and demands elaboration as does the apparent difference in p38γ. These observations, taken together with our published data, suggest a direct link to the liver metastatic process via conceptualization of a paraneoplastic hepatopathy, is likely.

## Figures and Tables

**Figure 1 vaccines-08-00720-f001:**
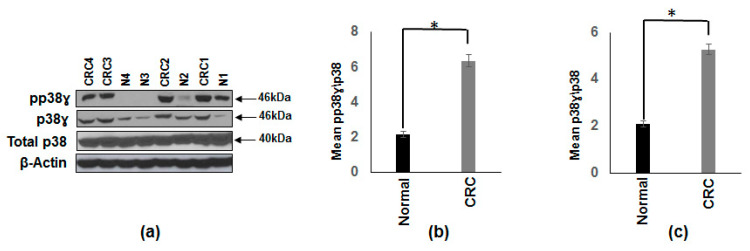
Increased protein expression and activation of p38γ in human cancerous colonic tissue compared to normal subjects. (**a**) Whole tissue extracts from colon cancer patients and normal subjects were prepared and 25 µg of protein subjected to SDS-gel electrophoresis and Western blot analysis using antibodies against p38γ and pp38γ. Equal loading was confirmed using an antibody against total p38 antibody. Representative blotting is shown for four normal (N) and four colon cancer (CRC) subjects. (**b**) Densitometric values expressed as fold increase of ratio of phosphorylated p38γ/total p38. (**c**) Densitometric values expressed as fold increase of the ratio of p38γ/total p38. The data was analyzed using the paired, two-tailed Student’s *t*-test, and the results were expressed as fold change ± SEM of 10 colon cancer patients and 10 normal subjects * *p*-value < 0.001.

**Figure 2 vaccines-08-00720-f002:**
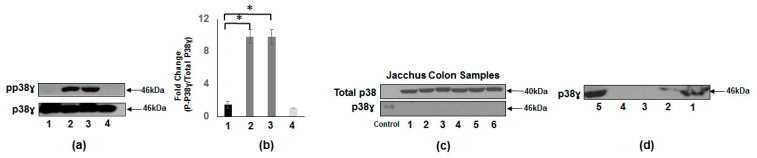
Presence of p38γ and pp38γ in preclinical and human tissue. (**a**) Whole tissue extracts from normal subject, CTT liver, and two derived from human cancerous colorectal tissue were prepared and subjected to SDS-PAGE and Western blot analysis using specific antibodies to pp38γ. Equal loading was confirmed using antibodies against p38γ. Normal subject (lane 1), CRC patients (lane 2, lane 3), and CTT liver (lane 4). (**b**) Densitometric values (means ± SEM) expressed as ratio of pp38γ/p38γ. * *p* value < 0.001. (**c**) Whole tissue extracts prepared from common marmoset (CM-*C. jacchus*) colon tissues (lanes 1–6) not at risk representing a negative control, in contrast CM small intestinal ileum are potentially at risk and a positive control (doxorubicin treated breast cancer cell line) were all similarly subjected to SDS-PAGE and Western blot analysis using specific antibodies to p38 and p38γ. None of the CM tissues have p38γ but all express total p38 (α and β) (*n* = 6). (**d**) Whole tissue extracts prepared from different organs of common marmoset, liver (lanes 1 and 2), gall bladder (lane 3), colon (lane 4), and ileum (lane 5) were subjected to SDS-PAGE and Western blot analysis using specific antibody to p38γ (*n* = 2).

**Figure 3 vaccines-08-00720-f003:**
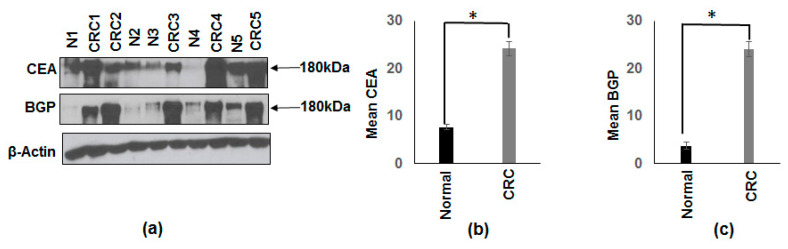
Increased protein expression of carcinoembryonic antigen (CEA) and biliary glycoprotein (BGP) from cancerous and corresponding normal tissues from patients with CRC were compared to normal colon from patients without colon cancer. (**a**) Whole tissue extracts from colon cancer (CRC) subjects and normal (N) subjects (included in this figure are four samples shown in Figure 1a with an additional sample making a total of five) were subjected to Western blot analysis using specific antibodies to CEA and BGP. Equal loading was confirmed with β-actin. A representative blot is shown for five normal subjects and five CRC subjects. (**b**) Densitometric values are expressed as fold increase of the ratio of CEA/β-actin (mean± SEM) from 10 CRC and 10 normal patients, * *p*-value < 0.007. (**c**) Densitometric values are expressed as a fold increase of the ratio of BGP/β-actin (mean± SEM) from 10 CRC and 10 normal patients, * *p*-value < 0.005. As shown in (**a**) the CRC patients exhibited increased protein expression of BGP and CEA compared to normal subjects.

**Figure 4 vaccines-08-00720-f004:**
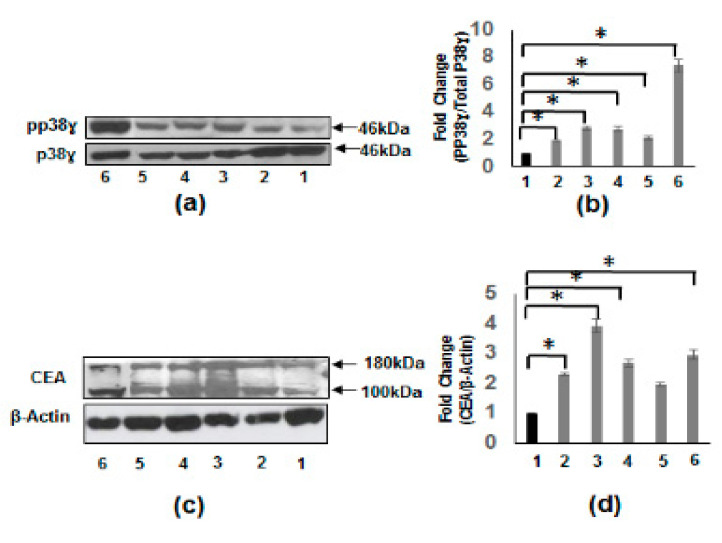
Increased phosphorylation of p38γ and protein expression of CEA in CRC cell lines. (**a**) Western blot using specific antibodies to p38γ and pp38γ in cell extracts from IEC-6 rat intestinal epithelial cells (lane 1), HCT 116 human CRC cells (lane 2), HT-29 human CRC cells (lane 3), Caco-2 human CRC cells (lane 4), SW620 human CRC cell line (lane 5), and HepG2 human hepatoblastoma carcinoma cell line (lane 6). (**b**) Densitometric values expressed as fold increase of the ratio of pp38γ/p38γ mean ± SEM (*n* = 4) * *p*-value < 0.05. The above results show the increased phosphorylation in CRC cell lines and HepG2 cells compared to normal IEC-6 cells. (**c**) Whole cell extracts from different cell lines were subjected to Western blotting with specific antibodies to CEA. (**d**) Densitometric values showing the fold increase of the ratio of CEA/β-actin mean ± SEM (*n* = 4) * *p*-value < 0.05. As shown in (c) the CEA levels are uniformly increased in colon cancer cell lines and HepG2 cell line.

**Figure 5 vaccines-08-00720-f005:**
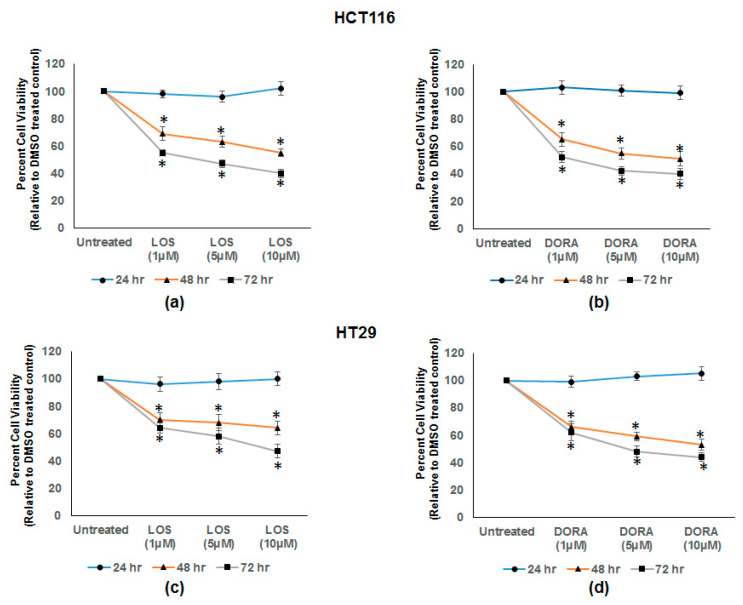
Cell growth inhibition of human CRC HT-29 and HCT 116 cell lines by losmapimod and doramapimod. HCT-116 and HT-29 CRC cells were treated with DMSO (untreated) or with different doses of losmapimod (LOS) and doramapimod (DORA) for 24, 48, and 72 h. Determination of viable/live cells was carried out by MTT assay as described in Section 2. (**a**,**b**) show the dose dependent inhibition of cell growth at 48 and 72 h by losmapimod and doramapimod in HCT-116 cells. Similarly, (**c**,**d**) show the dose dependent inhibition in HT-29 cells. The results are shown as mean ± SEM (*n* = 6); * *p*-value < 0.005. The dose is on the x-axis, and the viability of cells on y-axis.

**Figure 6 vaccines-08-00720-f006:**
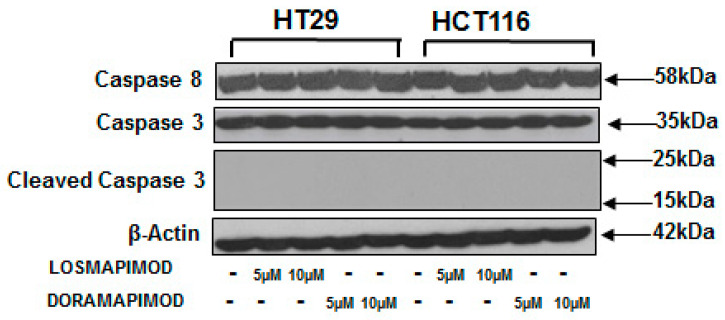
Inhibition of p38 MAPKs did not alter the caspase 8 or caspase 3 activity in CRC cells. HCT-116 and HT-29 CRC cell lines were treated with different doses of losmapimod or doramapimod or DMSO (control) for 48 h. Whole cell extracts were prepared and subjected to SDS-PAGE. Western blotting was performed using specific antibodies to caspase 8 and caspase 3. Equal loading was confirmed with β-actin. As shown above, neither caspase 3 or caspase 8 levels were altered by losmapimod or doramapimod.

**Figure 7 vaccines-08-00720-f007:**
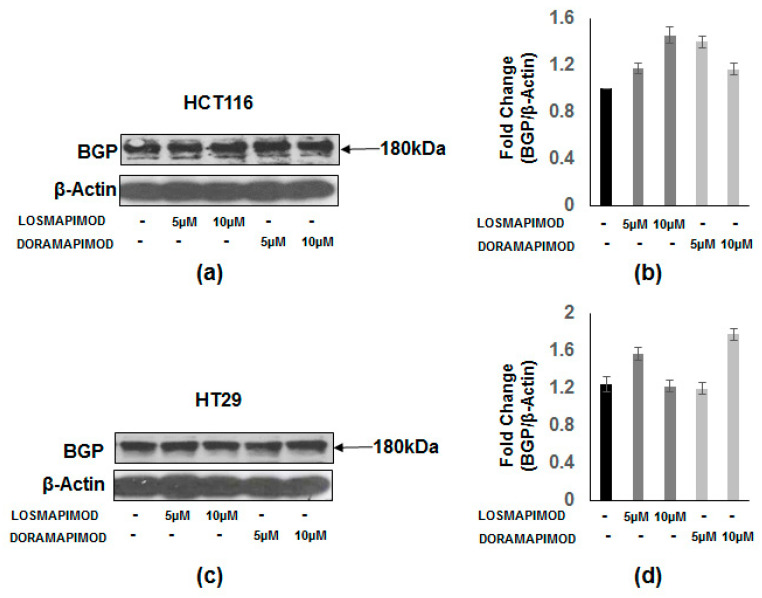
Lack of inhibition of biliary glycoprotein (BGP) protein levels by losmapimod or doramapimod by in colon cancer cells. (**a**,**b**) HCT-116 and (**c**,**d**) HT-29 CRC cells were treated with different doses of losmapimod or doramapimod or DMSO (control) for 48 h. Whole cell extracts were prepared and SDS-PAGE was performed followed by Western blotting with specific BGP antibodies. Equal loading was confirmed with β-actin. Panels (**b**,**d**) depict the resultant densitometric values showing fold change of the ratio of BGP/β-actin in HCT-116 and HT-29 cells.

**Figure 8 vaccines-08-00720-f008:**
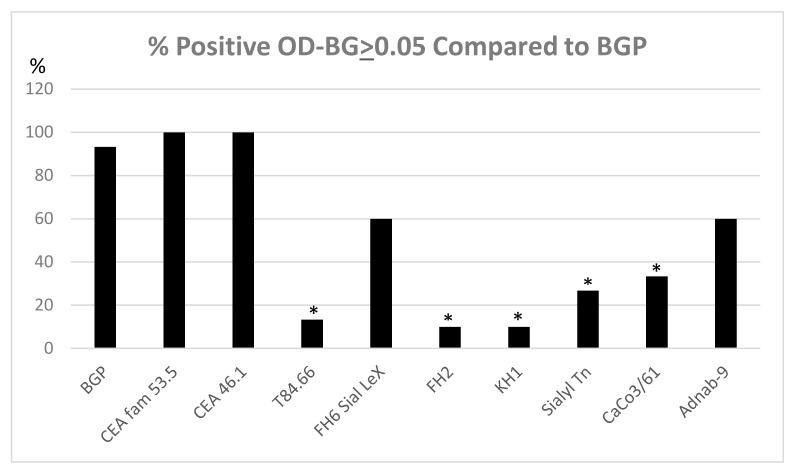
Expression of tumor metastasis associated antigens in tamarin and marmoset colonic tissue. Tissue extracts were obtained from the Callitrichidae family of nonhuman primates; cancerous (CTT *n* = 5) and corresponding normal mucosa (CTT *n* = 5) and (CM *n* = 5). The expression of tumor metastasis associated antigens were determined by ELISA. Antibody and antigen group designations: BGP-Biliary glycoprotein (CEACAM1), CEA Superfamily 53.5, CEA family 46.1, T84.66,FH6 Sial LeX fucoganglioside 6B, FH2 sialylated Lewis *X*, KH1 Extended Lewis Y blood group, Sialyl Tn α-sialyl Tn, CaCo3/61 fucosylated aminoproteoglycan, Adnab-9 Paneth cell glycoprotein p87. Positivity status was determined by an OD-background ≥0.05 as previously described [9]. Significant differences comparisons were made to BGP by nonparametric tests and the * indicates a *p* value < 0.002. Only CTT extracts were available for most of the blood group antigen testing. Additional comparisons to human can be found as described [34]

**Figure 9 vaccines-08-00720-f009:**
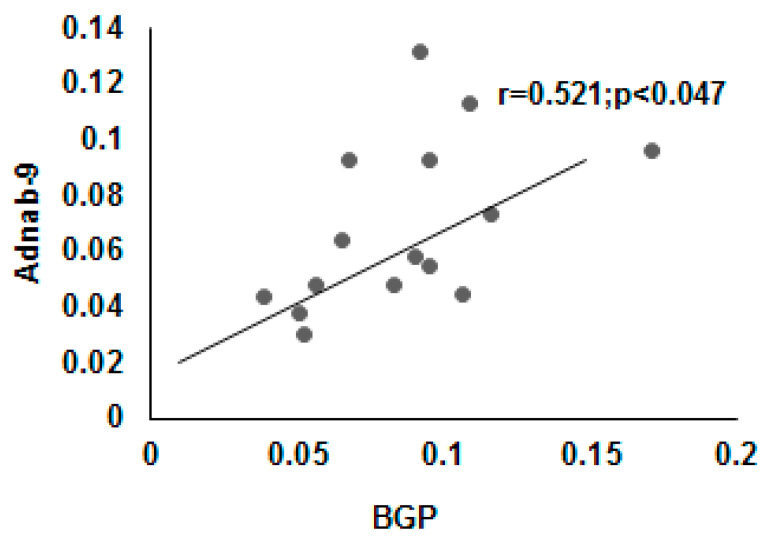
The correlation between BGP and Adnab-9 epitope expression in colonic tissues of CTT and CM. The scatter gram plot shows a statistically significant correlation between BGP antibody binding and Adnab-9 antibody binding by ELISA as previously published in [34].

**Figure 10 vaccines-08-00720-f010:**
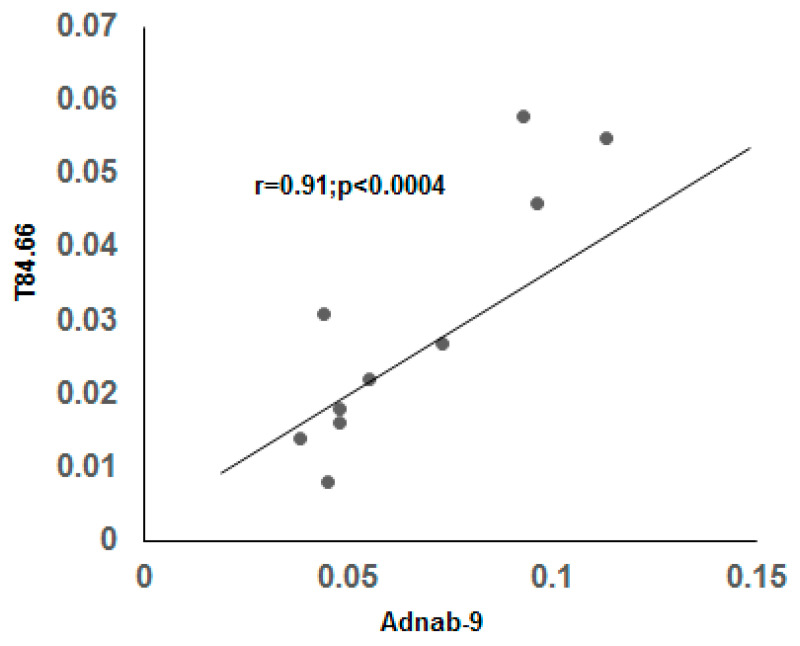
The correlation between Adnab-9 antibody binding and T84.66 antibody binding to CTT cancerous and normal tissue extracts. The scatter gram plot shows a statistically significant correlation between Adnab-9 and CEA. There was no correlation between Adnab-9 and anti-CEA T84.66 binding in CM tissues.

**Figure 11 vaccines-08-00720-f011:**
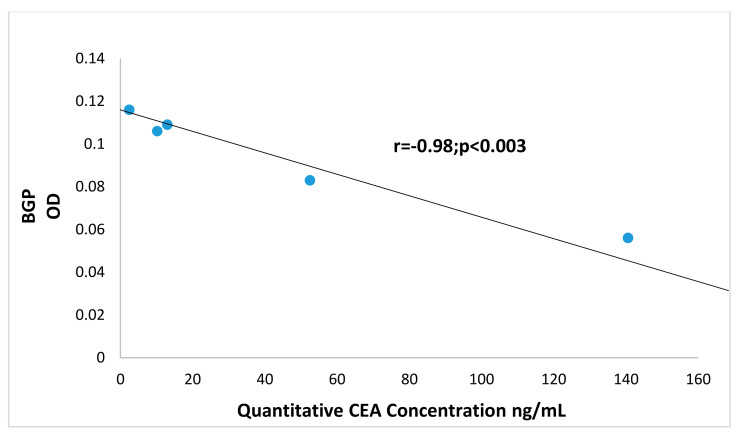
Normal CTT tissue correlation of BGP versus CEA in ng/mL shows an inverse relationship. The scatter gram plot shows a statistically significant correlation between BGP and CEA concentration. There was a similar correlation between BGP and CEA concentration when expressing the results in ng/mg protein in these normal-appearing mucosal sample CTT extracts. These extracts shown here are identical to the normal CTT shown in Figure 8, Figure 9 and Figure 10. There was no BGP versus CEA concentration correlation seen in the cancer tissue extracts. The CEA concentrations were obtained using the TOSOH Medics kit [25].

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
