# Peer review of "p38γ Activation and BGP (Biliary Glycoprotein) Induction in Primates at Risk for Inflammatory Bowel Disease and Colorectal Cancer—A Comparative Study with Humans"

_vaccines, 2020, doi:10.3390/vaccines8040720_

Round 1

Reviewer 1 Report

To the Authors: Your paper entitled "p38γ activation and BGP (biliary glycoprotein) induction in primates at risk for inflammatory bowel disease and colorectal cancer- A comparative study with humans." addresses the very important health concern on the topic of CRC that warrants further studies on this devastating disease.   The present study assessed p38γ activation in CRC patients, cancer cells, and tissues of cotton top tamarin (CTT) and common marmoset (CM). The results support the claim demonstrating that p38γ protein and phosphorylation levels are significantly increased in CRC patients compared to normal subjects as well as in human CRC cells but not hepatoblastoma cells or CM colon.  The data establish a reasonably convincing relationship that CEA and BGP are induced in the CRC patients that showed phosphorylation of p38γ thereby setting the stage for future investigations on the p38γ system in CRC. The following points, however, should be addressed:

1)  It is not clear why the journal Vaccines was selected for publication as this article is written in the vein of a targeted therapy rather than vaccine development.  Suggestion: publish in another MPDI journal

2) Figure legend in figure 3c says *p-value <0.007, *p-value <0.005.  the *p-value <0.007, may belong with panel b.  Suggestion: review the figure legend.

3) Section 3.5 on the inhibitors Losmapimod and Doramapimod plots show a similar trend suggesting the additional inhibition of gamma and delta don’t contribute much more than inhibiting both alpha and beta. Suggestion:  Don't over interpret the data.

4) Same concern on overinterpretation for data on BGP for experiments using Losmapimod and Doramapimod

5) Figure 8 x axis should be labeled 1-10.  Should have error bars as experiments should be done at least in triplicate.

6)  The comment: "The use of inhibitors to p38 that have variable specificities demonstrated that both losmapimod (p38 ɑ  & β  inhibitor) and doramapimod (p38 ɑ , β , γ , and δ  inhibitor) significantly inhibited the cell growth of CRC cell lines in a specific dose-effect." is not supported by the data.  In other words a "dose-effect" was not supported by the data and that phrase should be removed.

7)  The most controversial claim is the discrepancy on the HEPG2 cell line as it is a direct affront to authors of publication #20.  It would be prudent to check the cell line and redo the assay for HEPG2 using the same antibody as was reported in the extended data figure 7b of reference #20.  Those authors list the catalog numbers for products used.

Author Response

Comments and Suggestions for Authors

To the Authors: Your paper entitled "p38γ activation and BGP (biliary glycoprotein) induction in primates at risk for inflammatory bowel disease and colorectal cancer- A comparative study with humans." addresses the very important health concern on the topic of CRC that warrants further studies on this devastating disease.   The present study assessed p38γ activation in CRC patients, cancer cells, and tissues of cotton top tamarin (CTT) and common marmoset (CM). The results support the claim demonstrating that p38γ protein and phosphorylation levels are significantly increased in CRC patients compared to normal subjects as well as in human CRC cells but not hepatoblastoma cells or CM colon.  The data establish a reasonably convincing relationship that CEA and BGP are induced in the CRC patients that showed phosphorylation of p38γ thereby setting the stage for future investigations on the p38γ system in CRC. The following points, however, should be addressed:

  • It is not clear why the journal Vaccines was selected for publication as this article is written in the vein of a targeted therapy rather than vaccine development. Suggestion: publish in another MPDI journal

Response to Reviewers of manuscript entitled:”p38γ activation and BGP (biliary glycoprotein) induction in primates at risk for inflammatory bowel disease and colorectal cancer- A comparative study with humans”, by H. Talwar, B. McVicker and M. Tobi.

Please see the line by line rebuttal as requested.  We are extremely grateful and indebted to the reviewer’s for their thorough and thoughtful reviews and helpful constructive criticism.  In response to the review we have made major changes, adding and correcting both text and figures which we believe has greatly enhanced the clarity and readability of our paper.  The changes are detailed below.

Reply to reviewer#1

The following paragraph and references have been added to the discussion to justify the Journal selection. We were invited by the Journal to submit our work for their special issue dedicated to primate research.

Both the CTT and more recently the CM have been used as models for vaccination research, superior to the older macaque model (1,2). By focusing on elements of the MAPK and an innate-immune system effector cell product we believe that we have provided a novel perspective that combines these elements that could be integrated into the newly advocated holistic approach to vaccine development (3). In addition, since the CM is an animal model for serious coronavirus infection (4), this may have research implications for COVID-19 vaccine and therapeutics development. Given these considerations, and that the MAPK are active in innate cell immunity (5), we believe that our research is most relevant and may potentially inform future vaccine development. See lines 442-449.

  1. Finerty S, Machett M, Arrand JR, Watkins PE, Tarlton J, Morgan AJ. Immunizations of cotton top tamarins and rabbits with a candidate vaccine against the Ebsyein-Barr virus based on the major viral envelope glycoprotein gp340 and alum. Vaccine 1994 13:1180-4, doi: 10.1016/0264-410x(94)90240-2.
  2. Smith DR, Johnston SC, Piper A, Botto IM, Donnelly G, Shamblin J, Albarino CG, Hensley LE, Schmaljohn C, Nichol ST, Burd BH. Attenuation and efficacy of live-attenuated Rift Valley fever virus vaccine candidates in non-human primates. PLoS Negl Trop Dis 12(5):e0006474. https://doi.org/10.1371/journal.pnid.0006474 [doi.org].
  3. van den Boorn JG, Hartmann G. Turning tumors into vaccines: co-opting the innate immune system. Cancer Immunol Immunother DOI: 10.1016/j.immuni.2013.07.011 [doi.org].
  4. Falzarano D. de Wit E, Feldmann F, Rasmussen Al, Okumura A, et al. 2014. Infection with
    MERS-CoV causes lethal pneumonia in the common marmoset. PLoS 10(8):e1004250.doi:10.1371/jpurnal.ppat.1004250.
  5. Umasuthan N, Bathige SD, Noh JK, Lee J. MAPK kinases active in innate cell immunity.  Gene structure, molecular characterization and transcriptional expression of two p38 isoforms (MAPK11 and MAPK14) from rock bream (Oplegnathus fasciatus). Fish Shellfish Immunol. 2015 Nov;47(1):331-43. doi: 10.1016/j.fsi.2015.09.018. Epub 2015 Sep 10. PMID: 26363230. See lines 626-645.  

2) Figure legend in figure 3c says *p-value <0.007, *p-value <0.005.  the *p-value <0.007, may belong with panel b.  Suggestion: review the figure legend. Please see correction on lines 210-212.

3) Section 3.5 on the inhibitors Losmapimod and Doramapimod plots show a similar trend suggesting the additional inhibition of gamma and delta don’t contribute much more than inhibiting both alpha and beta. Suggestion:  Don't over interpret the data. Please see the corrections lines 241 & 398.

4) Same concern on overinterpretation for data on BGP for experiments using Losmapimod and Doramapimod. We only mentioned that these inhibitors show no effect on BGP levels in both CRC cell lines. Please see the corrections ( lines 269-270).

5) Figure 8 x axis should be labeled 1-10.  Should have error bars as experiments should be done at least in triplicate.

Rebuttal to Critique 5. This has been corrected and we are glad that this was brought to our attention by the reviewer. The results depicted by the bars represent the proportions of tests that were positive with each antibody. The testing is by non-parametric testing. This was intended to inform the reader of the relative positive expression of other antigens relative to BGP. We have also corrected the data as for the most of blood group antigens there were only sufficient CTT tissue extract for n=10. See lines 298-307.    

6)  The comment: "The use of inhibitors to p38 that have variable specificities demonstrated that both losmapimod (p38 ɑ  & β  inhibitor) and doramapimod (p38 ɑ , β , γ , and δ  inhibitor) significantly inhibited the cell growth of CRC cell lines in a specific dose-effect." is not supported by the data.  In other words a "dose-effect" was not supported by the data and that phrase should be removed. Please see the corrections lines 241 & 398.

7)  The most controversial claim is the discrepancy on the HEPG2 cell line as it is a direct affront to authors of publication #20.  It would be prudent to check the cell line and redo the assay for HEPG2 using the same antibody as was reported in the extended data figure 7b of reference #20.  Those authors list the catalog numbers for products used. In fact we tested the same antibody used in reference #20 (ɣp38 MAPK from Cell Signaling). But we are working on another western blot using HepG2 cell line from another source and will provide you the results very soon. We really thankful to the reviewer for providing us this critical suggestion.

Reviewer 2 Report

Comment to the authors:   In this study, the authors tried to determine the role of CEACAMs and p38-gamma in CRC patients.  However, this paper is not well stratified and very confusing. I think this paper needs whole reconsideration and professional proofreading. At least, the author should clear the aim of the study in the introduction. In the result section, they need to explain why each experiment is needed to accomplish the aim of the study. Finally, in the discussion, the author can summarize the result in the present study in relation to the aim of the study. Other information could be attached if it helps an easy understanding of readers.   Major comments:
  • In Figure1a, the authors represent western blotting using 4 CRC samples. However, the authors use 5 CRC samples in Figure 3a. Are they independent samples? 
  • Figure 2b compares p38-gamma expression between rat liver and human CRC patients. But why use rats?  Rat samples could not be the correct controls, because their rat could be different from humans in the expression pattern of any proteins. Please use healthy human tissues in comparison.
  • In Figure 2c/d, the author showed p38 expression in the common marmoset, but what is the aim of the experiments? Especially, the authors should describe why this experiment is necessary to accomplish the aim of the study.
  • In Figure 4, the authors compared rat cells and human cells. The difference could be derived from the difference between species, rather than the CRC condition. Please use some human intestinal epithelial cells or their cell-line as control. 
  • In Figure 8, the author should show the lane number on the x-axis. And what the “Positive” means on the y-axis? Ratio of positive cells in 10 (5 CTT + 5 CM) samples? If so, they could be separated by species, because different species could be different expression profiles. And it is unclear why the authors need to assess the expression of markers in other primates. It should show in human samples. 
  • In Figure 9. Adnab-9 could be the name of the monoclonal antibody, so "Adnab-9 expression" is incorrect,  "Adnab-9 epitope expression" seems to be better. The author should discuss the meaning of the expression in detail. Also, they need to describe how to detect the expression in the materials and methods.
  •  
  Minor comments:
  • In Figure 5, it could be easy to understand that the time (24, 48, and 72 hr) is on the x-axis, and the dose is on the plot.
  • in the title, “Inflammatory bowel disease” is not fit the theme of the study, please reconsider it. 
  • In the discussion (L357), the authors described “we hypothysed …”, but this kind of hypothesis should be set before the beginning of the study. So, it should be explained in the introduction. Also, the contents in this paragraph could be described in the introduction.
  •  

Author Response

Comments and Suggestions for Authors

Comment to the authors:   In this study, the authors tried to determine the role of CEACAMs and p38-gamma in CRC patients.  See line 43 However, this paper is not well stratified and very confusing. I think this paper needs whole reconsideration and professional proofreading. At least, the author should clear the aim of the study in the introduction. In the result section, they need to explain why each experiment is needed to accomplish the aim of the study. Finally, in the discussion, the author can summarize the result in the present study in relation to the aim of the study. Other information could be attached if it helps an easy understanding of readers.   Major comments:

Response to Reviewers of manuscript entitled:”p38γ activation and BGP (biliary glycoprotein) induction in primates at risk for inflammatory bowel disease and colorectal cancer- A comparative study with humans”, by H. Talwar, B. McVicker and M. Tobi.

Please see the line by line rebuttal as requested.  We are extremely grateful and indebted to the reviewer’s for their thorough and thoughtful reviews and helpful constructive criticism.  In response to the review we have made major changes, adding and correcting both text and figures which we believe has greatly enhanced the clarity and readability of our paper.  The changes are detailed below.

  • In Figure1a, the authors represent western blotting using 4 CRC samples. However, the authors use 5 CRC samples in Figure 3a. Are they independent samples? See line 205/207.
  • Figure 2b compares p38-gamma expression between rat liver and human CRC patients. But why use rats?  Rat samples could not be the correct controls, because their rat could be different from humans in the expression pattern of any proteins. Please use healthy human tissues in comparison. Please find attached the new amended blot 2 a & b with healthy normal subject. Please see lines 101-106.
  • In Figure 2c/d, the author showed p38 expression in the common marmoset, but what is the aim of the experiments? Especially, the authors should describe why this experiment is necessary to accomplish the aim of the study. See lines 184-7.
  • In Figure 4, the authors compared rat cells and human cells. The difference could be derived from the difference between species, rather than the CRC condition. Please use some human intestinal epithelial cells or their cell-line as control. See lines 101-103.
  • In Figure 8, the author should show the lane number on the x-axis (This has been corrected putting in the recognized epitopes). And what the “Positive” means on the y-axis. This was explained in the foot note OD-background >0.05 but now we have added the word status to better define positivity (Line 301). Ratio of positive cells in 10 (5 CTT + 5 CM) samples? If so, they could be separated by species, because different species could be different expression profiles. Seeing that antibody profiles are similar adding these to the figure might be confusing but we did check for differences in means between the species and none were significant-See lines 319-320. And it is unclear why the authors need to assess the expression of markers in other primates. It should show in human samples. Many  of these comparisons have been previously published please see reference 35 See line 302/303 
  • In Figure 9. Adnab-9 could be the name of the monoclonal antibody, so "Adnab-9 expression" is incorrect,  "Adnab-9 epitope expression" seems to be better. This will be added in the graphics note. See line 320.The author should discuss the meaning of the expression in detail. Also, they need to describe how to detect the expression in the materials and methods. This was added See line 322.
  •  

  Minor comments:

  • In Figure 5, it could be easy to understand that the time (24, 48, and 72 hr) is on the x-axis, and the dose is on the plot. We would prefer to leave as is because we want to emphasize the dosage element. Corrected: See line 250-251.
  • in the title, “Inflammatory bowel disease” is not fit the theme of the study, please reconsider it. The CTT develops cancer as a result of inflammatory bowel disease and the CM also suffers from this malady but the cancers are in the small bowel. IBD is the inalienable background on which this animal scenario plays out and to omit it would be to eliminate the biological context. We agree however that this should be discussed. Please see lines 335-337.
  • In the discussion (L357), the authors described “we hypothysed …”, but this kind of hypothesis should be set before the beginning of the study. So, it should be explained in the introduction. Also, the contents in this paragraph could be described in the introduction. This was entered into the introduction. Line 44-45.

Reviewer 3 Report

The authors present new findings that p38γ activation and high expression of CEA and BGP are induced in CRC patients and these characteristics could be the risk for colorectal diseases. However, some of the results supporting the conclusions are unclear.

Specific suggestions:

  1. In Figure 2a, the authors show that pp38γ is detected only in the human tissues. However, there remains the possibility that the antibody does not work in CTT tissues. Because there is a concern about the reliability of the anti-pp38γ antibody, the authors should examine and confirm cross-reactivity of the antibody used in study. 

  1. Several immunoblots are not clear. As an example, p38γ immunoblot in Figure 2d is unclear. Especially, multiple bands are shown in the blots of CEA in Figure 4c. It seems to CEA is expressed only in HepG2 cells (lane 6, 180kDa). The authors should present convincing data.

Author Response

Comments and Suggestions for Authors

The authors present new findings that p38γ activation and high expression of CEA and BGP are induced in CRC patients and these characteristics could be the risk for colorectal diseases. However, some of the results supporting the conclusions are unclear.

Response to Reviewers of manuscript entitled:”p38γ activation and BGP (biliary glycoprotein) induction in primates at risk for inflammatory bowel disease and colorectal cancer- A comparative study with humans”, by H. Talwar, B. McVicker and M. Tobi.

Please see the line by line rebuttal as requested.  We are extremely grateful and indebted to the reviewer’s for their thorough and thoughtful reviews and helpful constructive criticism.  In response to the review we have made major changes, adding and correcting both text and figures which we believe has greatly enhanced the clarity and readability of our paper.  The changes are detailed below.

Specific suggestions:

  1. In Figure 2a, the authors show that pp38γ is detected only in the human tissues. However, there remains the possibility that the antibody does not work in CTT tissues. Because there is a concern about the reliability of the anti-pp38γ antibody, the authors should examine and confirm cross-reactivity of the antibody used in study. Additional proof by our immunoprecipitation data. In order to prove the validity of ɣpp38 antibody, we immunoprecipitated the lysates from  rat liver, CRC subject and CTT with ɣp38 MAPK antibody and then probed with ɣpp38 antibody on a western blot. We got only phospho band with CRC subject (data not shown).

  1. Several immunoblots are not clear. As an example, p38γ immunoblot in Figure 2d is unclear. Especially, multiple bands are shown in the blots of CEA in Figure 4c. It seems to CEA is expressed only in HepG2 cells (lane 6, 180kDa). The authors should present convincing data.. The weak band that is in HepG2 cells is not at 180kDa but is just above the 100kDa. We have replaced a better picture showing the 180kDa band.

Round 2

Reviewer 2 Report

I have no additional comments.
